# LC-MS Based Phytochemical Profiling towards the Identification of Antioxidant Markers in Some Endemic *Aloe* Species from Mascarene Islands

**DOI:** 10.3390/antiox12010050

**Published:** 2022-12-26

**Authors:** Célia Breaud, Laura Lallemand, Gary Mares, Fathi Mabrouki, Myriam Bertolotti, Charlotte Simmler, Stéphane Greff, Morgane Mauduit, Gaëtan Herbette, Eldar Garayev, Christophe Lavergne, Maya Cesari, Sok-Siya Bun-Llopet, Béatrice Baghdikian, Elnur Garayev

**Affiliations:** 1Aix Marseille Univ, CNRS 7263, IRD 237, Avignon Université, IMBE, 27 Blvd Jean Moulin, Service of Pharmacognosy, Faculty of Pharmacy, 13385 Marseille, France; 2CYROI, Plateforme de Recherche, Cyclotron Réunion Océan Indien, 97490 Saint-Denis, France; 3CNRS, Centrale Marseille, FSCM, Spectropole, Aix Marseille Université, Campus de St Jérôme-Service 511, 13397 Marseille, France; 4Department of General and Toxicological Chemistry, Azerbaijan Medical University, Baku AZ1001, Azerbaijan; 5CBNM Conservatoire Botanique National de Mascarin, 2, rue du Père Georges, Les Colimaçons, 97436 Saint-Leu, France

**Keywords:** *Aloe* spp., La Reunion, molecular network, chemotaxonomy, LC-MS, antioxidant

## Abstract

*Aloe* plant species have been used for centuries in traditional medicine and are reported to be an important source of natural products. However, despite the large number of species within the *Aloe* genus, only a few have been investigated chemotaxonomically. A Molecular Network approach was used to highlight the different chemical classes characterizing the leaves of five *Aloe* species: *Aloe macra, Aloe vera, Aloe tormentorii*, *Aloe ferox*, and *Aloe purpurea*. *Aloe macra, A. tormentorii*, and *A. purpurea* are endemic from the Mascarene Islands comprising Reunion, Mauritius, and Rodrigues. UHPLC-MS/MS analysis followed by a dereplication process allowed the characterization of 93 metabolites. The newly developed MolNotator algorithm was usedfor molecular networking and allowed a better exploration of the *Aloe* metabolome chemodiversity. The five species appeared rich in polyphenols (anthracene derivatives, flavonoids, phenolic acids). Therefore, the total phenolic content and antioxidant activity of the five species were evaluated, and a DPPH-On-Line-HPLC assay was used to determine the metabolites responsible for the radical scavenging activity. The use of computational tools allowed a better description of the comparative phytochemical profiling of five *Aloe* species, which showed differences in their metabolite composition, both qualitative and quantitative. Moreover, the molecular network approach combined with the On-Line-HPLC assay allowed the identification of 9 metabolites responsible for the antioxidant activity. Two of them, aloeresin A and coumaroylaloesin, could be the principal metabolites responsible for the activity. From 374 metabolites calculated by MolNator, 93 could be characterized. Therefore, the *Aloe* species can be a rich source of new chemical structures that need to be discovered.

## 1. Introduction

The *Aloe* genus counts over 500 species, mainly distributed in arid areas, predominantly in Africa, but also in India [1]. *Aloe* species are members of the Asphodelaceae family, and the studied species belong to the same subfamily Alooidae. *Aloe vera* (L.) Burm. f. (synonym *A. barbadensis* Mill.) and *Aloe ferox* Mill. (synonym *Cape Aloe*), are two of the most well-known species. Within the Alooidae subfamily, the Aloe section *Lomatophyllum* (Willd.) G.D. Rowley includes species such as *Aloe tormentorii* (Marais) L.E. Newton and G.D. Rowley, *Aloe macra* Haw. and *Aloe purpurea* Lam., which are endemic to the Mascarene Islands.

In the Mascarene Islands, leaves of *Aloe* species are used for their cutaneous healing properties [2]. *Aloe* leaves contain two different parts: the gel and the leaf exudate. The gel produced by *Aloe* leaves is frequently used in traditional medicine to treat skin injuries, including wounds and sunburns. This gel, found in the inner part of the leaf, contains polysaccharides and glycoproteins, considered to be involved in wound healing activity [3]. The bitter leaf exudate is known for its laxative properties. The leaves are also known for their antispasmodic effect and for relieving discomfort associated with menstruation [3,4].

Amongst the pharmacological properties described for *Aloe* species preparations, the antioxidant properties are the most frequently evaluated and mainly attributed to the ethanol extracts obtained from the leaves of *Aloe vera.* Such properties play an important role in the protection against oxidative stresses caused by free radicals and are involved in multiple inflammatory diseases [1]. Antioxidants are mostly supplied by fruits and vegetables. Some of them are polyphenolic derivatives. However, when dealing with complex mixtures, such as plant extracts, it is often challenging to target the compounds responsible for the antioxidant activity. A few single bioactive antioxidants, such as ascorbic acid, most commonly known as vitamin C, have been identified and additionally demonstrated to reduce the risk of coronary heart disease and cancer [5].

Several studies, such as the study conducted by Lobine et al. in 2017, described the phytochemical composition of 5 *Aloe* species, *A. tormentorii*, *A. purpurea*, *A. macra*, *A. lomatophylloides* and *A. vera* from the Mascarene Islands [2,6]. However, these studies used targeted LC-MS analyses allowing the annotation of only 21 metabolites. The recent use of the MS-based Molecular Network approach can facilitate structural dereplication and accelerate the annotation of new structural entities in complex samples. Molecular Networking is a bioinformatic tool enabling the visualization of non-targeted tandem mass spectrometry data (MS/MS). It has proven to be a very efficient tool to identify new metabolites in complex mixtures and is now hugely used in the field of Natural Product chemistry (NP), with the introduction of online platforms such as Global Natural Products Social Molecular Networking (GNPS) created by Wang et al. in 2016 [7,8].

Although studies have shown some insights into the chemical classification of *Aloe* plants, most of the species of this genus remain largely unexplored; *A. vera* and *A. ferox* remain the most known and used species.

Our work aims to comprehensively describe the phytochemical composition of five *Aloe* species sampled in Reunion Island: *A. macra, A. vera*, *A. tormentorii*, *A. ferox*, and *A. purpurea,* using untargeted MS-based molecular networking analyses. In parallel, an On-Line HPLC DPPH method was developed to facilitate the identification of antioxidant phytochemicals. The antioxidant activity and the total polyphenols content of the five *Aloe* leaf extracts were, thus, comparatively evaluated.

## 2. Materials and Methods

### 2.1. Plant Material

Leaf samples of *Aloe tormentorii* (MAZ 18) (GPS coordinates: −20.906363637386843, 55.49723509705913) species were collected on the 13th of March 2021, at the CIRAD Station de la Bretagne in St-Denis, Reunion Island.

Leaf samples of the *Aloe macra* (MAZ 16) (GPS coordinates: −21.13800553663294, 55.29488144618424), *A. purpurea* (MAZ 21) (GPS coordinates: −21.137998031426875, 55.29482243758612) and *A. ferox* (MAZ 19) (GPS coordinates: −21.137754112018204, 55.29680190783811) species were collected the 25th of August 2021, at the Conservatoire Botanique National de Mascarin in St-Leu, Reunion Island. A leaf sample of *Aloe vera* (MAZ 17) (GPS coordinates: −21.037980428963056, 55.217169537787626) species was collected on the same day at the 5 bis rue des Sables, in St-Paul, Reunion Island.

Voucher specimens were registered in the Herbarium of Reunion Island University with the respective barcodes: *A. tormentorii* REU025151, *A. macra* REU025140, *A. purpurea* REU025142, *A. ferox* REU025149, and *A. vera* REU025152.

All samples were cut into pieces and freeze-dried for 72 h (COSMOS 20 K, Cryotec, France). The materials were then crushed into powder using a knife mill (cutter mill) and packed in 50-mL tubes for transport.

### 2.2. Metabolite Extraction and Sample Preparation

One gram of the samples was weighed and extracted in 20 mL of 70% ethanol (1:20 *m/v*), using ultrasound-assisted extraction for 15 min at 25 °C (PEX05 25 kHz, Reus France). At 7.5 min, the crude extracts were agitated for 30 s using a vortex (VWR mixer mini vortex, EU). The resulting solutions were filtered twice: first under vacuum using glass sintered filters (Redisep 25 g 15–45 µm filters), then using 0.22 µm PTFE filters (Restek, France) into glass vials. Three analytical replicates were prepared for each species and stored at −20 °C until analysis.

The remaining filtrates of 15 mL were dried using a Speedvac (Thermo Scientific Savant Speedvac Concentrator SPD131DDA, equipped with a Thermo Scientific Savant Refrigerated Vapor Trap RVT5105 and an Edwards Pump RV8), then freeze-dried (Cryotec, France). The dried extracts were stored in air-tight containers at ambient temperature in the dark.

### 2.3. UHPLC-MS/MS Analysis

The LC-MS/MS analysis was performed on a Thermo Scientific Dionex 3000 Ultra High-Performance Liquid Chromatography system (UHPLC) coupled to a Bruker Impact II Q-TOF high-resolution mass spectrometer equipped with an electrospray ionization source (ESI). The chromatographic separation was carried on an Agilent Zorbax Eclipse Plus C18 column (2.1 × 100 mm, 1.8 µm) at 43 °C. Ultrapure water (A) (LC-MS grade, Carlo Erba, Italy) and acetonitrile (B) (LC-MS grade, Carlo Erba, Italy), both acidified with 0.1% formic acid (LC-MS grade, Carlo Erba, Italy), were used as mobile phases. The injection volume was 1 µL for all samples, and the flow rate of the mobile phase was 0.8 mL/min. The following gradient was applied: isocratic hold at 5% B for 2 min, 5–50% B over 2–17 min, 50–100% B over 17–27 min, then isocratic hold at 100% B for 2 min (27–29 min), followed by a decrease to 5% B in 1 min (29–30 min), held at 5% B over 30–33 min for the column equilibration for the next experiment.

Mass spectrometry data were acquired from *m/z* 50 to 1200, using both positive (+) and negative (-) modes. The following parameters were used for the Q-TOF in both ionization modes: end plate offset at 500 V; nebulizer gas (N_2_) pressure at 3.5 bar; dry gas flow (N_2_) at 12 L/min; drying temperature at 200 °C; acquisition rate at 4 Hz. The capillary voltage was set at 3500 V for positive mode and 3000 V for negative mode.

A data-dependent acquisition (DDA) protocol was used; therefore, MS/MS fragmentation spectra were obtained automatically for the three most abundant precursor ions using mixed collision energy 20–40 eV (in stepping mode).

A solution of Sodium Formate Acetate was used as a calibration to obtain high mass accuracy (2–5 ppm) and was automatically injected at the beginning of each run.

Five commercial standards were solubilized in MeOH at a concentration of 1 mg/mL and injected with the same method as the hydro-ethanolic extracts to confirm annotations. The following standards were injected: chlorogenic acid (5-*O*-caffeoylquinic acid) (Extrasynthese, 4991S, lot.327-97-9), isoorientin (Extrasynthese 1055S, lot.08030310), isovitexin (Extrasynthese 1235S, lot.98052204), vitexin (Extrasynthese 1232S, lot.0142511), aloin A (Sigma Aldrich, lot.085K1111) and loliolide. The loliolide standard was purified in our previous study, and the identification has been confirmed by MS and NMR data [9].

### 2.4. Data Processing and Molecular Network

#### 2.4.1. File Conversion

Raw datasets obtained from the UHPLC-MS/MS system were calibrated using Bruker DataAnalysis (5.0 SR1 64-bit) and converted into open format .mzXML using GNPS Vendor 32-bit [8].

However, the issue regarding a non-calibrated precursor ion value in .mzXML, highlighted by Zdouc et al. in 2021, persists in Bruker DataAnalysis software [10], which means that the export of raw data sets results in calibrated MS/MS data, with non-calibrated precursor ion information (<precursorMz></precursorMz>) for each MS/MS scan. Zdouc et al. suggested using a script in DataAnalysis software to export to .mgf files, which contain calibrated precursor ion information, and a Perl5 script to correct precursor information in the .mzXML data. The Perl5 script could not be used in the present work because of the difference between Bruker Compass versions used for the acquisition of raw data. Therefore, the Perl5 script was adapted to be used with Bruker Compass version 5.0 SR1 64-bit. It was also adjusted to correct the data exported with GNPS Vendor in 32 and 64 bits. The script is freely available on https://github.com/elnurgar/dataanalysis.git (accessed on 13 November 2022).

#### 2.4.2. Data Processing

Exported .mzXML data were pre-processed using MZmine software, version 3.0. The processing workflow includes raw data file import, mass detection, chromatogram building, chromatogram deconvolution, feature list deisotoping, alignment between analytical replicates, filtering, and plant species [8,11,12]. The features present in blank methanol runs were removed from the features list. Parameters of each step used for processing can be seen in Appendix A of Appendix A.

Processed data were exported (mgf and CSV files) and applied to MolNotator software. First, the AdNotator module allows the identification of the adducts of a compound, and FragNotator identifies the in-source fragments. Then, the MolNet module allows the construction of a molecular network [13]. The Adnotator precursor and fragments ion mass tolerance were set to 0.002 and the retention time tolerance to 6 s. Detailed parameters used for MolNotator software are presented in the Appendix A. Cytoscape 3.9.1 software was used to visualize the resulting network.

The table with 374 metabolites and their peak areas in the studied *Aloe* species, generated by MolNotator, was submitted on Metaboanalyst 5.0 platform for hierarchical cluster analysis in order to produce a dendrogram illustrating molecular similarity between five *Aloe* species [14].

### 2.5. Total Phenolics Content (TPC)

The Folin-Ciocalteu method was adapted from El Hosry et al. [15]. The previously dried hydro-ethanolic extracts were prepared at 3 mg/mL in EtOH 50% (*v*/*v*). A volume of 5 mL of the prepared solutions was mixed with 1 mL of Folin-Ciocalteu reagent (Sigma Aldrich, lot BCBP2077V) and 4 mL of Na_2_CO_3_ 7.5% (m/v) (Fluka Biochemika, 347579/1 596 lot.71345) in a 100 mL volumetric flask. The volume was completed with distilled water. Samples were incubated in an oven for 2h30 at 30 °C in the dark. The absorbance of the solutions was measured at 760 nm using a UV/Vis spectrophotometer (Thermo Scientific Genesys 10S UV-Vis). TPC was expressed as g of gallic acid equivalent (GAE) per 100 g of extracts.

### 2.6. Evaluation of Antioxidant Activity

#### 2.6.1. DPPH Assay in 96-Well Plates

The DPPH assay was realized according to Blois et al. and adapted for a 96-well plate [16]. The dried hydro-ethanolic extracts were solubilized in EtOH 70% (*v*/*v*) and diluted at different concentrations, which were optimized to reach the EC_50_: *A. macra* (40–250 µg/mL), *A. vera* (200–3000 µg/mL), *A. ferox* (500–3000 µg/mL), *A. tormentorii* (40–250 µg/mL), *A. purpurea* (40–250 µg/mL).

Gallic acid (Extrasynthese, 6079 lot.04900102), used as a positive control, was solubilized in MeOH and diluted to obtain concentrations in the range of 0.5–5 μg/mL.

A fresh DPPH methanolic solution at a concentration of 10^−4^ M was prepared every day by dissolving 10 mg of DDPH (Sigma Aldrich, D9132-5G lot.STBD2362V) in 250 mL MeOH and kept at room temperature in the dark for 3 h before use.

DPPH assay was carried out in 96-well plates (Sterilin Ltd., Newport, UK) with one blank row, one negative control row, three columns with sample solutions at different concentrations and in triplicates and one column of sample blank solution. The 96-well plate layout can be seen in Figure 1. The composition of each solution was:

Blank: 250 μL of methanol (MeOH)

Negative control: 50 μL of MeOH and 200 μL of DPPH 40 mg/L MeOH

Sample or positive control: 50 μL of the sample or positive control solution and 200 μL of DPPH solution

Sample or positive control blank: 50 μL of each sample or positive control and 200 μL of MeOH

The 96-well plate was placed in the spectrophotometer (BioTek EON, Providence, RI, USA) and was incubated for 1 h at 25 °C. Absorbance was then read at 515 nm. The scavenging activity (%) was expressed as EC_50_ (concentration corresponding to 50% inhibition).

Statistical analysis was performed by ordinary one-way ANOVA test followed by Dunnett’s multiple comparisons tests.

#### 2.6.2. On-Line RP-HPLC-DPPH

A rapid On-Line method allowing targeting compounds with radical scavenging activity in complex mixtures was realized according to Koleva et al. [17]. A scheme of the online system is given in Figure 2. The HPLC Agilent 1260 system coupled with Agilent 1200 consisted of the following:Agilent 1260: a sample injector system (vial sampler G7129A); a HPLC pump delivery system (binary pump G7112B); a column oven (MCT G7116A); a DAD UV detector (DAD G7117C)Agilent 1200: a second HPLC pump (quaternary pump G1311A) for the delivery of the DPPH solution; a DAD UV-Vis detector (DAD G1315B).

Chromatographic separation was carried out on an Agilent Zorbax Eclipse Plus C18 column (2.1 × 100 mm, 1.8 µm). The reaction coil was a 10 m × 0.25 mm i.d. stainless steel tube. The UV detection wavelength for the tested samples was set at 325 nm. Detection of DPPH solution bleaching was carried out at 515 nm.

Dried hydro-ethanolic extracts were prepared at a concentration of 10 mg/mL in EtOH 70% (*v*/*v*), then filtered using 0.22 µm PTFE filters into glass vials.

A fresh DPPH methanolic solution at a concentration of 2 × 10^−4^ M was prepared every day by dissolving 40 mg of DDPH in 500 mL MeOH and kept at room temperature in the dark for a minimum of 3 h before use. This allowed the stabilization of the solution absorbance.

All solvents used were of HPLC grade. Ultrapure water (A) and acetonitrile (B) (Carlo Erba, Italy), both acidified with 0.1% formic acid (Carlo Erba, Italy), were used as the mobile phase. The injection volume of all samples was 2 µL, and separation was carried at 43 °C, with a mobile phase flow of 0.2 mL/min. The following gradient was applied: isocratic hold at 5% B for 2 min, 5–50% B over 2–17 min, 50–100% B in 1 min (17–18 min), then isocratic hold at 100% B for 2 min (18–20 min), followed by a decrease to 5% B in 0.1 min (20–20.1 min), held at 5% B over 20.1–30 min for the column’s equilibration for the next experiment.

The second HPLC pump delivering the DPPH at 2 × 10^−4^ M in methanol solution into the reaction coil was set at a flow of 0.2 mL/min. The reaction coil temperature was set at 60 °C. The used flow rate and the reaction coil dimensions allowed a reaction time of 1 min 14 s between samples and the DDPH solution.

LC-MS/MS analysis was carried out using the same analytical method as On-Line DPPH for the annotation of peaks presenting radical scavenging activity. The mass spectrometer parameters used were the same as in Section 2.3.

## 3. Results and Discussion

### 3.1. Molecular Network and Chemotaxonomic Study

The use of a Molecular Network approach was chosen to explore the phytochemical composition of the five *Aloe* species. The leaves of five *Aloe* species collected at Reunion Island have been extracted with 70% ethanol. Metabolomics spectral data from the hydro-ethanolic leaf extracts of the *Aloe* species were acquired in both ionization modes (ESI +/−) using an LC-MS quadrupole time-of-flight (Q-TOF).

Raw data sets were converted into open .mzXML format and processed using MZmine. The final features lists contained 1370 features in positive mode and 1596 in negative mode. The use of MolNotator allowed us to identify the different adducts generated during the ionization process using a triangulation method and predict the neutral metabolites [13]. The algorithm also allowed the identification of in-source fragments with high efficiency, therefore decreasing the number of false positives. The MolNotator algorithm proposed 374 metabolites out of the 2966 chemical features.

Based on the list of the 374 metabolites, a hierarchical clustering analysis was generated using the Metaboanalyst platform and can be found in Figure 3A. The MolNotator molecular network was generated based on cosine similarity between different MS/MS spectra of predicted metabolites (Figure 3B).

The visualization of the metabolome through a molecular network approach revealed structurally related chemical families in the five *Aloe* species. Visual exploration of the network showed that MS/MS spectra were grouped according to their chemical class.

The automatized dereplication process combined with manual annotation allowed for the putative and full identification of 94 metabolites, including metabolites so far undescribed in the *Aloe* genus. In addition, chemotaxonomic characteristics reported for the *Aloe* genus helped to improve the structural annotation process [6,18].

Identified compounds were classified following the levels of confidence proposed by Schymanski et al.: level 1 (L1): structure confirmed by the reference standard with MS, MS/MS spectra, and retention time matching; level 2a (L2a): probable structure using library spectrum match or literature match; level 2b (L2b): diagnostic of structure using MS/MS fragments or ionization behavior, with no literature confirmation; level 3 (L3): tentative candidates with uncertainties (for example positional isomers) [19].

A total of six reference standard compounds were injected for MS/MS data acquisition to confirm the identification of major metabolites: chlorogenic acid (8), isoorientin (31), isovitexin (33), loliolide (34), vitexin (41) and aloin A (61).

All annotated metabolites can be found in Table 1. Spectral data acquired in both positive and negative ionization modes were used to annotate the *Aloe* species metabolomes.

#### 3.1.1. Major Chemical Classes of the Aloe Species Metabolome

Three main families of metabolites could be described in the five *Aloe* species.

Firstly, a range of phenylpropanoids (C6-C3), more precisely cinnamic acid derivatives, such as chlorogenic acids, were observed in the extracts and can easily be recognized by the presence of a characteristic caffeoyl substituent (fragment ion at *m*/*z* 163.0390 in positive mode). Likewise, coumaroylquinic acid and feruloylquinic acids and their derivatives presented a fragment ion at *m*/*z* 147.0440 and *m*/*z* 177.0550 in positive mode. They are all strongly linked with cosine scores above 0.95, suggesting their similar fragmentation patterns.

Coumaroylquinic acid and its derivatives are ubiquitous, as they were found in all five *Aloe* species. The three naturally occurring isomers of caffeoylquinic acid were observed in many *Aloe* species [26,38]. Chlorogenic acid (5-*O*-caffeoylquinic acid) (8) was identified by comparing its retention time and MS/MS spectrum with a commercial standard. The two other mono-caffeoylquinic acid isomers were also detected (5; 13) and annotated through their characteristic MS/MS fragmentation patterns [23]. The distribution of described isomers was found to be species-dependent: chlorogenic acid (8) was found only in *A. vera*, while the two other isomers, neochlorogenic acid (3-*O*-caffeoylquinic acid) (5) and cryptochlorogenic acid (4-*O*-caffeoylquinic acid) (12) were mostly found in *A. ferox, A. macra*, and *A. purpurea.* As opposed to the other investigated *Aloe* species, *A. tormentorii* contained few cinnamic acid derivatives: only a glycosylated derivative of coumaric acid was annotated (9).

Secondly, the five studied species were rich in flavonoids, which are biosynthesized through the phenylpropanoid pathway. Various subclasses can be found in plants due to the action of reductases, isomerases, dioxygenases, and hydrolases. Thus, flavonoids can be found with structurally diverse aglycone backbones, namely chalcones, flavones, isoflavones, flavanols, flavonols, flavanones, and anthocyanidins. These backbones exist in various modified forms through hydroxylation, methylation, and glycosylation by transferases.

The loss of a sugar attachment represents the main MS fragmentation path for flavonoids: a neutral loss of 162.0528 amu represents a loss of a hexose, while the neutral loss of 132.119 characterizes a pentose.

In the studied *Aloe* species, flavones were the most frequent subclass of flavonoids: apigenin and luteolin were found with diverse sugar attachments and in many isomeric forms. C-glycosyl flavones, such as isoorientin and isovitexin, showed to be especially present. The second subclass that could be found was flavonols, such as kaempferol. No free flavone or flavonol aglycones were detected in the studied *Aloe* species.

Lastly, several lipids were identified: phospholipids, more precisely lysophosphatidylcholine (LPC) derivatives, were found in all five species. LPC, also called lysolecithins, is a class of lipids resulting from the cleavage of phosphatidylcholine (PC) via the action of phospholipase A_2_ (PLA_2_) and/or the transfer of fatty acids to free cholesterol via lecithin-cholesterol acyltransferase (LCAT).

Two fatty acids, linoleic acid (92) and α-linolenic acid (85) were also annotated in all studied species and are widely described in plants as they contribute to the integrity of the cellular membrane [39].

Pheophorbide A (92), a product of chlorophyll breakdown, was also detected [40].

#### 3.1.2. Chemotaxonomic Exploration of the Chemical Composition of the Five Aloe Species

From 374 metabolites found, 241 metabolites (64.4%) were common to two or more species, and 133 metabolites (35.6%) were specific to a species.

Among the metabolites present in all studied *Aloe* species, major compounds previously described in the *Aloe* genus could be highlighted: aloin A (61) and B (58), which are members of the anthracene chemical group [41]. Aloesin (10) and aloeresin A (68), members of the chromone family, were also detected. These four metabolites are largely described in the literature as specific to the *Aloe* genus [6,22,42].

*Aloe macra* presented the richest chemical diversity, as it contains 219 metabolites in total, 26 being specific. *Aloe purpurea* presented the second chemical diversity with 209 metabolites in total. However, only 19 of them were specific to *A. purpurea*. *Aloe vera* contains 190 metabolites, but 55 were uniquely found in the species. Amongst the five studies species, *A. vera* showed the most specific metabolites. *Aloe ferox* contains 163 metabolites, 27 specific to the species. *Aloe tormentorii* appeared to have the lowest metabolic diversity, with 157 metabolites and only six specific ones.

The highest specificity of metabolites observed for *A. vera* and *A. ferox* can be explained by the fact that these species do not belong to the *Lomatophyllum* section. These results show that Mascarene *Aloe* species, *A. purpurea*, *A. tormentorii*, and *A. macra* possess a different metabolome compared to other species of the genus *Aloe*.

Hierarchical clustering analysis, generated by the Metaboanalyst platform (Figure 3A), demonstrated the differences between the metabolic fingerprints of *A. vera* and *A. ferox* compared to those characterizing the *Aloe* species uniquely found in the Mascarene islands. Within the Mascarene *Aloe* species, the metabolomic fingerprints were closer between *A. tormentorii* and *A. purpurea* compared to *A. macra*, which is endemic to Reunion Island [43]. These results are also in agreement with the study from Ranghoo-Sanmukhiya et al. [2], where genetic similarities were determined between the same Mascarene *Aloe* species: *A. purpurea*, *A. tormentorii, A. macra* compared to *A. vera*. The authors showed that *A. purpurea* and *A. tormentorii* share more genetic similarities with *A. macra* than *A. vera*. Herein, we corroborated such classification from a metabolomic point of view.

A detailed exploration of the molecular network highlighted that the major and most abundant metabolites of the *Aloe* genus could be found at the center of the main cluster. Aloin A (61) and B (58), aloesin (10), and aloeresin (coumaroylaloesin) (68) form the main cluster with their derivatives.

A number of malonylnataloin derivatives were also detected and can be seen at the top of the network. Only two of them seemed to be specific to *A. purpurea*.

One of the known major metabolites, aloesin (10), was detected in all five species. Interestingly, some metabolites in the aloesin derivatives cluster were mostly detected in *A. macra*.

*Aloe vera* appeared richer in coumaroylaloesin derivatives, which are almost entirely specific to this species (except for one metabolite that could also be found in *A. ferox*). Two other chromones, isoaloeresin D (59) and isorabaichromone (55), were only present in *A. vera*.

It appears that the main cluster links a range of different chemical classes, but all belong to the phenolic class. Phenolic acids, and more precisely, cinnamic acid derivatives, can be found in the middle of the cluster, between aloesin and aloin derivatives. They form the denser part of the cluster, with a great number of nodes, and are strongly linked to Aloin and its derivatives.

Flavonols and flavones can easily be identified on the network, as they form two subclusters. Visualization of the chemical space of the five *Aloe* species through a molecular network pointed out that *A. macra* contain a larger number of metabolites, while *A. vera* contains a lower diversity of metabolites, but more specific ones: for example, flavonols such as kaempferol-3-glucoside, also named astragalin (51), were found strictly in *A. vera*, and kaempferol-3-*O*-rutinoside (48) was found in *A. vera*, with traces in *A. ferox*.

Flavones were distributed across all five studied species. However, apigenin-*C*-hexoside-*O*-hexoside (28) was found only in *A. vera*, and narcissin or narcissoside (52) only in *A. ferox*. This can indicate the presence of a specific flavonoid biosynthesis pathway for *A. vera.*

*Aloe tormentorii* showed no specific subcluster and appeared to contain less diverse metabolites than the four other species.

Lipids identified through annotation were not found as part of a cluster on the network and are distributed among the single nodes.

### 3.2. Total Phenolic Content (TPC)

From the 93 annotated metabolites, more than 2/3 belong to the polyphenols class. Therefore, the total phenolic content was evaluated for hydro-ethanolic extracts of the *Aloe* species.

The total phenolic contents of the five species, expressed in gallic acid equivalent, were determined according to the Folin-Ciocalteu method [15]. Phenolic contents ranged from 1.1 to 2.8 g GAE/100 g extract (Table 2), with the highest TPC for *A. purpurea*.

Higher TPC is often correlated with higher radical scavenging activity, which was evaluated in this study using the DPPH assay.

### 3.3. DPPH Assay

#### 3.3.1. DPPH Assay in 96-Well Plates

This method was developed by Blois to determine the antioxidant activity of compounds using a stable free radical α,α-diphenyl-β-picrylhydrazyl (DPPH) [16]. The assay measures the scavenging capacity of antioxidant compounds towards DPPH. The single electron of the DPPH’s nitrogen atom is reduced by receiving a hydrogen atom from the scavenging compound, forming the corresponding hydrazine [44].

The 96-well plate assay was realized as a preliminary step to determine whether or not extracts possessed a DPPH scavenging activity. EC_50_ is defined here as the concentration of substrate that causes a 50% reduction in the DPPH absorbance at 515 nm. The lower the EC_50_, the higher the scavenging activity. In our study, the extracts with the EC_50_ below 200 µg/mL were considered active. Three extracts showed antioxidant activity: *A. macra* (EC_50_ = 172 µg/mL), *A. ferox* (EC_50_ = 151 µg/mL), with the best result obtained from *A. purpurea* hydro-ethanolic extract with an EC_50_ of 88 µg/mL. The extracts of *A. vera* and *A. tormentorii* with an EC_50_ superior to 200 µg/mL were considered inactive. The statistical analysis shows that results are significant for all *Aloe* extracts with a *p*-value ≤ 0.05, excluding the results for *Aloe purpurea* extract.

It is to be noted that results between TPC and DPPH assays were not directly correlated: this can be explained by the diversity in the polyphenolic compounds composition of the extracts, the presence of groups on phenolics that can interfere with the colorimetric reaction of TPC assay, or the presence of compounds in extracts that may also act as false positives. However, two species, *A. macra* and *A. purpurea*, with the highest metabolite diversity, were richest in TPC, with relatively high antioxidant DPPH activity. The same tendency regarding the activity of Mascarene *Aloe* species is observed in the previous study conducted by Govinden-Soulange et al., where *A. macra* (from Reunion Island) and *A. purpurea* were more active than *A. vera* and *A. tormentorii* [45]. In order to determine which metabolites contribute the most to the radical scavenging activity, the 96-well plates DPPH assay was followed by the On-Line RP HPLC DPPH assay.

#### 3.3.2. On-Line RP HPLC DPPH Assay

The method described by Koleva et al. can be applied to complex mixtures such as plant extracts and/or fractions for rapid detection of radical scavenging components [17].

Such a method was applied to hydro-ethanolic extracts of all five studied species in triplicates. Combined UV and DPPH bleaching Visible (Vis) chromatograms of active extracts can be seen in Figure 4. The chromatograms of inactive extracts are presented in Appendix A. *Aloe vera* and *A. tormentorii*, both inactive extracts, showed no decrease in absorbance on the Vis chromatogram, which means that none of the separated compounds within the extracts induced bleaching of the DPPH solution. This could mean that (1) compounds present in *A. vera* and *A. tormentorii* possess low to no antioxidant properties, (2) their concentration in the extracts is too low to be effective, and (3) compounds act antagonistically as antioxidants.

On the other hand, the extracts that had antioxidant activity in 96-well plates also showed radical scavenging properties with the On-Line approach. *A. macra, A. ferox*, and *A. purpurea* showed 5 to 6 negative peaks on the DPPH Vis 515 nm chromatogram.

The aim of the On-Line RP HPLC DPPH assay was to identify compounds responsible for the antioxidant properties of the extracts.

The extracts were also analyzed by LC-MS/MS using the same analytical method as On-Line DPPH. The acquired MS data led to the annotation of active metabolites by comparing their MS/MS spectra to those from previously annotated metabolites (Section 2.1).

That way, aloesin (10), aloeresin A (2-*O*-*p*-coumaroylaloesin) (68), 2″-*O*-trans-*p*-coumaroylaloenin (59), two caffeoylquinic acid isomers (5; 12), luteolin-*C*-glucoside-*O*-pentoside (30), isoorientin (31) and one new compound, yet to be annotated, and codified, were identified as metabolites responsible for the radical scavenging properties. This new metabolite was found to be uniquely detected in *A. macra* and *A. purpurea*, underlining the potential of these species for the discovery of new bioactive compounds. However, most of the metabolites identified as responsible for the activity were not specific to the active species, namely *A. purpurea*, *A. ferox*, and *A. macra*, as they were present in all five species.

The quantitative factor may play an important role in the antioxidant properties of an extract. Our hypothesis is that metabolites identified as responsible for the radical scavenging properties can be found in different amounts and proportions across the five studied species. In order to confirm that the proportion of antioxidant compounds in active extracts is higher than in the other extracts, their peak heights were compared. The results, presented in Table 3, show a good correlation between the peak height of an antioxidant compound detected at 325 nm and the activity of the extracts. Even though some compounds responsible for the radical scavenging activity are present in inactive extracts, their content is too low to induce a decrease in the DPPH chromatogram baseline at 515 nm.

The main advantage of the On-Line RP HPLC DPPH assay is that it can guide the identification of radical scavenging molecules within a complex mixture such as a crude extract. Using this method, aloeresin A was found to be comparatively most abundant in the most bioactive extracts, namely *A. macra* and *A. purpurea*, whereas coumaroylaloesin was most abundant in *A. ferox*, which antioxidant activity was closer to the one measured for *A. macra* extract. Both compounds could be the principal responsible for the measured antioxidant activities.

By this approach, the compounds responsible for the biological activity can be highlighted without wasting time on the blind purification process of each compound for offline assays.

## 4. Conclusions

The use of computational tools to investigate the chemical composition of the leaves of the five *Aloe* species provided abundant information regarding the composition of specialized/secondary metabolites. The molecular network approach allowed a better view of the chemical diversity and the specificity of each species, as the five studied species showed phytochemical differences. Among the 374 metabolites calculated by the MolNotator algorithm, 241 metabolites were common to two or more species, and 133 were specific to one species. The endemic Mascarene *Aloe* species (*A. macra*, *A. tormentorii*, and *A. purpurea*) showed a molecular specificity compared to *A. vera* and *A. ferox.* The molecular network allowed the annotation of 93 metabolites, with some of them undescribed in the *Aloe* genus. Therefore, the *Aloe* species are the source of new bioactive compounds, as at least 281 metabolites still need to be discovered.

Moreover, the combination of chemotaxonomic study with DPPH On-Line assays led to the identification of 9 metabolites responsible for the antioxidant activity, such as isoorientin, aloeresin A, coumaroylaloesin and caffeoylquinic acids, belonging to phenolic acids, flavonoids and chromone derivatives chemical families. Two metabolites, aloeresin A and coumaroylaloesin, found to be most abundant in active extracts, could be mainly responsible for the radical scavenging activity.

These results emphasize the chemical diversity between *Aloe* species and the potential of the *Aloe* genus as a source of new bioactive agents.

## Figures and Tables

**Figure 1 antioxidants-12-00050-f001:**
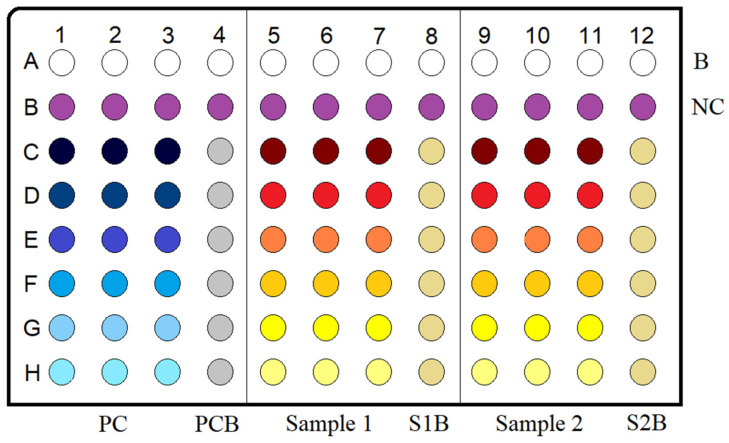
DPPH antioxidant assay in 96-well plate layout. B = blank, NC = negative control, PC = positive control, PCB = positive control blank, S1B = sample 1 blank, S2B = sample 2 blank. All samples were tested in triplicate.

**Figure 2 antioxidants-12-00050-f002:**
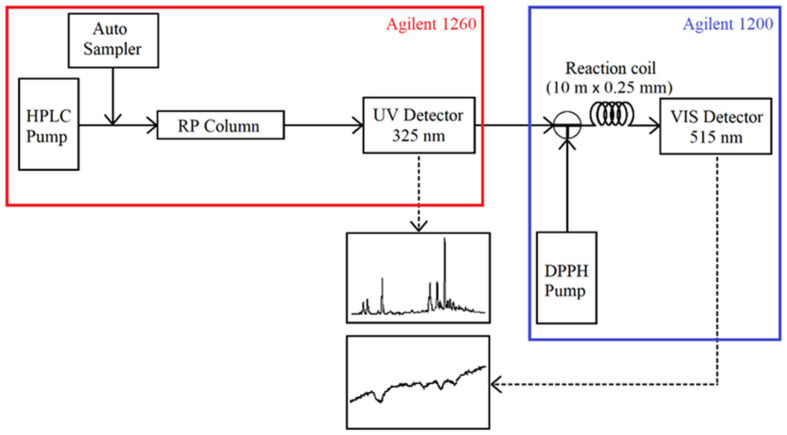
Instrumental setup for the On-Line RP HPLC DPPH radical scavenging assay.

**Figure 3 antioxidants-12-00050-f003:**
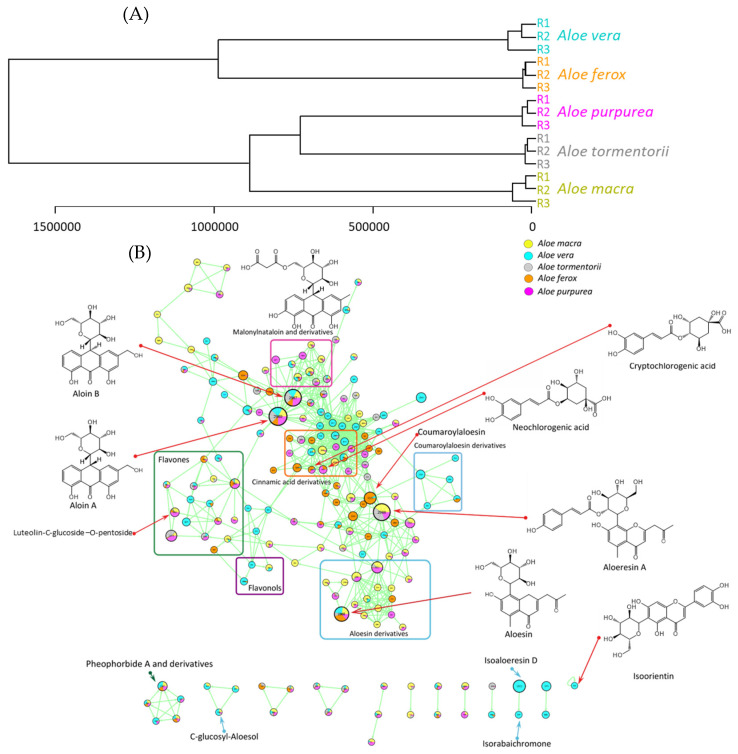
(**A**) Dendrogram illustrating molecular similarity among the five *Aloe* species calculated from 374 metabolites and generated with Metaboanalyst based on Euclidean distances and Ward clustering. (**B**) Molecular Network of the five *Aloe* species, analyzed by UHPLC-MS/MS using electrospray ionization in both modes (positive and negative), with six majors subclusters: flavones, flavonols, cinnamic acid derivatives, aloin derivatives, Aloeresin (coumaroylaloesin) derivatives, and Aloesin derivatives. Each node is calculated by triangulation based on its adducts and represents a molecule. Node colors represent the distribution across the five species (in terms of MS intensity), with the following codes: *Aloe macra* (yellow), *Aloe vera* (blue), *Aloe tormentorii* (grey), *Aloe ferox* (orange), *Aloe purpurea* (pink).

**Figure 4 antioxidants-12-00050-f004:**
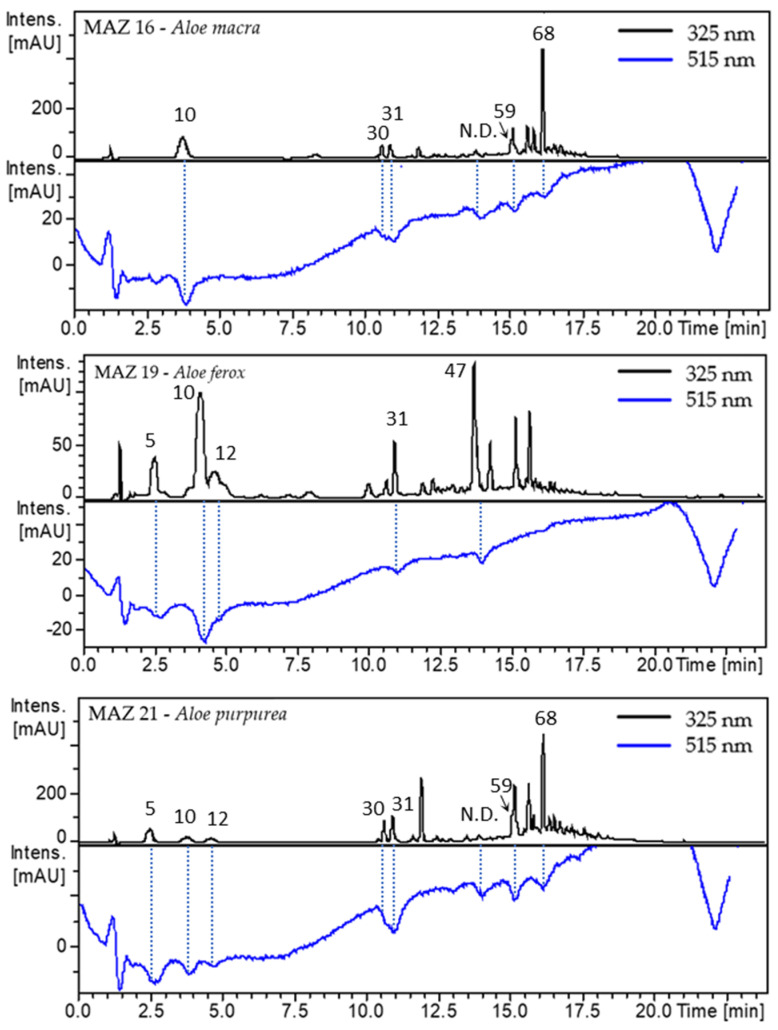
DPPH inhibitory activity of active extracts. Chemical structures of active compounds can be seen in Figure 3B. N.D. = Non determined; 5 = Neochlorogenic acid (3-*O*-caffeoylquinic acid); 10 = Aloesin; 12 = Cryptochlorogenic acid (4-*O*-caffeoylquinic acid); 30 = Luteolin-*C*-glucoside-*O*-pentoside; 31 = Isoorientin; N.D. = *m*/*z* [M+H]^+^ = 949.2767 (C_47_H_48_O_21_); 47 = Coumaroylaloesin; 59 = 2″-*O*-trans-*p*-coumaroylaloenin; 68 = Aloeresin A (2-*O*-*p*-coumaroylaloesin).

**Table 1 antioxidants-12-00050-t001:** Annotation of metabolites from the hydro-ethanolic extracts of the five *Aloe* species by liquid chromatography quadrupole time-of-flight mass spectrometry (LC-Q-TOF-MS) analysis in positive and negative ion modes. Metabolites are sorted by retention times (RT).

No	MN ID	Annotation	RT (min)	Molecular Formula	IC	MS				MS/MS		Ref	Presence				
						MM ID ^+^	[M + H]^+^ (Error in ppm)	MM ID ^−^	[M − H]^-^ (Error in ppm)	[M + H]^+^ Fragments (Relative Intensity in %)	[M − H]^-^ Fragments (Relative Intensity in %)		*Aloe*				
													M	V	T	F	P
1	3121	Succinoadenosine	1.11	C_14_H_17_N_5_O_8_	L2a	67	384.1150 (+0.1)	58	382.1007 (+0.7)	252.0728 (100); 162.0765 (25); 192.0514 (15); 136.0615 (14)	134.0478 (100); 206.0679 (33); 267.0949 (10)	[20]					✓
2	3123	Xanthurenic acid	1.21	C_10_H_7_NO_4_	L2a	76	206.0449 (+0.6)	69	204.0302 (−0.2)	178.0499 (100); 105.0334 (20); 206.0450 (15); 150.0546 (13)	160.0396 (100); 159.0334 (81); 131.0375 (47)	[20]	✓	✓	✓	✓	✓
3	3087	Ethyl citrate	1.37	C_8_H_12_O_7_	L2b	90	221.0657 (+0.5)	83	219.0511 (+0.3)	101.0233 (100); 83.0127 (65); 129.0548 (54); 139.0025 (17); 111.0079 (16)	111.0089 (100); 72.9934 (32); 154.9986 (10); 99.0085 (10); 85.0299 (10); 129.0196 (5)	[21]	✓	✓	✓	✓	✓
4	3128	Caffeoylquinic acid-hexoside	1.59	C_25_H_24_O_12_	L2b	106	517.1552 (+40)	98	515.1406 (+40)	163.0390 (100); 145.0278 (3); 135.0439 (2); 127.0386 (2)	191.0565 (100); 179.0357 (38); 353.0893 (17); 192.0598 (12); 341.0895 (11); 135.0454 (10);	[22]				✓	
5	2977	Neochlorogenic acid (3-CQA)	1.63	C_16_H_18_O_9_	L2a	113	355.1024 (+0.1)	106	353.0880 (+0.5)	163.0388 (100); 135.0439 (15); 145.0281 (9); 117.0334 (5)	191.0560 (100); 135.0448 (76); 179.0349 (60); 85.0292 (6);	[23]	✓			✓	✓
6	1726	Methylthioadenosine	1.92	C_11_H_15_N_5_O_3_S	L2a	131	298.0970 (+0.5)	ND	ND	136.0618 (100)	ND	[20]			✓	✓	
7	2967	Coumaroylquinic acid	2.73	C_16_H_18_O_8_	L2a	164	339.1076 (+0.5)	168	337.0929 (−0.1)	147.0442 (100); 119.0493 (23)	163.0400 (100); 119.0502 (77); 191.0560 (39); 173.0454 (6)	[6]	✓				✓
8	2969	Chlorogenic acid (5-CQA)	3.54	C_16_H_18_O_9_	L1	196	355.1023 (−0.2)	207	353.0879 (+0.3)	163.0390 (100); 135.0437 (12); 145.0286 (8); 117.0334 (4)	191.0562 (100); 85.0296 (4); 127.0403 (2)	[23,24]		✓			
9	2972	Coumaric acid glucoside	3.79	C_15_H_18_O_8_	L2b	209	327.1079 (+1.4)	223	325.0930 (−0.3)	147.0440 (100); 165.0547 (57); 119.0493 (17); 91.0548 (5)	145.0294 (100); 117.0345 (41); 163.0395 (4)				✓		
10	2968	Aloesin or Neoaloesin A	4.17	C_19_H_22_O_9_	L2a	230	395.1337 (+0.1)	ND	ND	233.0809 (100); 275.0915 (97); 395.1336 (73); 245.0810 (48); 203.0705 (34); 299.0916 (29)	ND	[6]	✓	✓	✓	✓	✓
11	3150	3-*O*-Feruloylquinic acid	4.25	C_17_H_20_O_9_	L2a	233	369.1182 (+0.5)	249	367.1035 (+0.1)	177.0549 (100); 145.0284 (39); 117.0334 (12); 149.0597 (5)	193.0506 (100); 134.0371 (91); 117.0346 (11); 149.0610 (7)	[22]	✓			✓	✓
12	3151	Cryptochlorogenic acid (4-CQA)	4.31	C_16_H_18_O_9_	L2a	237	355.1025 (+0.3)	263	353.0878 (−0.1)	163.0392 (100); 135.0442 (13); 145.0287 (9); 117.0336 (6); 89.0385 (3)	135.0450 (100); 173.0455 (97); 191.0559 (90); 179.0351 (74); 93.0344 (28); 85.0297 (9)	[23]				✓	✓
13	3159	C-glucosyl-(S)-aloesol	4.73	C_19_H_24_O_9_	L3	261	397.1493 (−0.1)	285	395.1346 (−0.4)	233.0804 (100); 203.0709 (50); 397.1503 (40); 277.1074 (30); 217.0856 (17); 243.1025 (14)	203.0717 (100); 231.0661 (72); 275.0925 (41); 351.1080 (29); 395.1351 (24); 247.0984 (9)	[25]	✓	✓	✓		✓
14	2986	Coumaroylquinic acid isomer	4.8	C_16_H_18_O_8_	L2a	277	339.1075 (+0.2)	298	337.0930 (+0.3)	147.0441 (100); 119.0494 (20); 91.0542 (6)	173.0454 (10); 119.0501 (31); 191.0565 (27); 93.0347 (22); 163.0403 (17); 137.0246 (7)	[6]	✓				
15	3164	2-Acetyl-5-hydroxy-3-methylphenyl β-D-glucopyranoside	4.92	C_15_H_20_O_8_	L2a	283	329.1230 (−0.3)	305	327.1088 (+0.8)	167.0705 (100); 149.0596 (11); 121.0657 (4)	165.0553 (100); 123.0462 (30); 149.0235 (18);	[20]		✓			
16	3046	Ethanone, 1-[4-(beta-D-glucopyra nosyloxy)-2,6-dihydroxyphenyl]-	5.10	C_14_H_18_O_9_	L2a	298	331.1024 (+0.1)	321	329.0879 (+0.3)	169.0497 (100); 151.0391 (11); 123.0443 (5); 127.0393 (3)	167.0351 (100); 123.0453 (48); 209.0454 (10); 191.0342 (10); 97.0310 (6); 146.9356 (6)	[20]		✓			
17	2966	Coumaroylquinic acid isomer	5.24	C_16_H_18_O_8_	L2a	313	339.1076 (+0.5)	339	337.0930 (+0.3)	147.0441 (100); 119.0489 (18); 91.0542 (5)	191.0564 (100); 93.0347 (25); 119.0503 (16); 163.0403 (15)	[6]	✓	✓		✓	
18	3009	Aloenin derivative	5.37	C_20_H_24_O_9_	L3	325	409.1491 (−0.5)	353	407.1350 (+0.6)	247.0965 (100); 409.1495 (90); 367.1393 (42); 217.9861 (37); 289.1075 (23); 233.0812 (18)	243.0666 (100); 407.1358 (54); 275.0934 (35); 191.0719 (19); 365.1254 (18); 215.0715 (18)			✓			
19	3010	Coumaroylquinic acid isomer	5.48	C_16_H_18_O_8_	L2a	339	339.1076 (+0.5)	371	337.0932 (+0.9)	147.0440 (100); 119.0490 (19); 91.0541 (5)	173.0457 (100); 93.0348 (26); 119.0503 (23); 163.0401 (23)	[6]	✓				✓
20	3172	Undulatoside A	5.57	C_16_H_18_O_9_	L2a	347	355.1023 (−0.2)	382	353.0879 (+0.3)	193.0498 (100); 191.0704 (35)	191.0336 (100); 75.0084 (69)	[20]		✓			
21	3084	Roseoside	5.62	C_19_H_30_O_8_	L2a	354	387.2014 (+0.1)	391	385.1869 (+0.3)	207.1384 (100); 95.0857 (69); 123.0807 (53); 149.0963 (28); 135.1172 (19); 113.0602 (18)	205.0506 (100); 190.0273 (68); 92.9971 (56)	[20]	✓				✓
22	3181	Lucenin II isomer	5.88	C_27_H_30_O_16_	L2a	393	611.1607 (+0.1)	418	609.1461 (−0.1)	329.0658 (100); 299.0550 (75); 353.0657 (58); 431.0983 (33); 395.0758 (33); 413.0867 (28)	609.1439 (100); 327.0502 (71); 447.0922 (45); 357.0613 (21)	[26]		✓		✓	
23	3184	Methyl aloesin	5.91	C_20_H_24_O_9_	L3	395	409.1495 (+0.5)	430	407.1351 (+0.8)	247.0964 (100); 289.1070 (92); 409.1495 (57); 259.0966 (52); 217.0856 (35); 313.1069 (28)	287.0929 (100); 407.1343 (65); 259.0973 (58); 217.0860 (29); 245.0817 (23); 317.1018 (9)		✓		✓		✓
24	3006	Feruloylquinic acid	6.12	C_17_H_20_O_9_	L2a	419	369.1180 (−0.1)	461	367.1036 (+0.4)	177.0550 (100); 145.0287 (29); 117.0339 (8); 149.0600 (5)	191.0557 (100); 134.0371 (25); 93.0343 (20); 173.0452 (18)	[22]	✓	✓		✓	✓
25	3038	Vicenin 2	6.23	C_27_H_30_O_15_	L2a	435	595.1652 (−0.9)	478	593.1511 (−0.2)	325.0705 (100); 379.9820 (84); 337.0709 (77); 355.0812 (61); 457.1136 (59); 439.1041 (54)	593.1508 (100); 353.0668 (55); 383.0768 (38); 473.1088 (27); 503.1191 (8); 527.0844 (5)	[26]		✓			
26	2973	Coumaroylquinic acid isomer	6.29	C_16_H_18_O_8_	L2a	440	339.1074 (−0.1)	486	337.0929 (+0.1)	147.0440 (100); 119.0491 (20); 91.0542 (6)	191.0560 (100); 85.0296 (6); 127.0399 (2)	[6]	✓	✓			
27	3065	Pectolinarin	6.44	C_29_H_34_O_15_	L3	465	623.1966 (−0.7)	ND	ND	299.0915 (100); 461.1436 (11)	ND		✓	✓			
28	3027	Apigenin-*C*-hexoside-*O*-hexoside	6.61	C_27_H_30_O_15_	L2b	485	595.1653 (−0.8)	532	593.1513 (+0.2)	313.0704 (100); 283.0599 (78); 337.0703 (56); 415.1020 (37); 397.0913 (33); 379.0812 (29)	593.1500 (100); 311.0557 (45); 297.0398 (22); 431.0978 (17); 473.1073 (7); 3441.0659 (5)	[27]		✓			
29	3062	Lucenin II	6.61	C_27_H_30_O_16_	L2a	487	611.1609 (+0.4)	536	609.1460 (−0.2)	329.0657 (100); 299.0549 (42); 353.0658 (38); 449.1075 (31); 413.0855 (17); 395.0763 (15)	609.1453 (100); 489.1033 (43); 429.0822 (32); 327.0513 (32); 298.0487 (23); 369.0608 (21)	[26]	✓			✓	✓
30	2992	Luteolin-*C*-glucoside-*O*-pentoside	6.70	C_26_H_28_O_15_	L2b	494	581.1503 (+0.3)	548	579.1356 (+0.1)	329.0659 (100); 449.1082 (74); 299.0554 (56); 353.0660 (45); 431.0973 (22); 413.0874 (21)	579.1356 (100); 459.0935 (43); 298.0482 (38); 309.0404 (19); 327.0510 (16); 429.0829 (14)	[20]	✓	✓	✓	✓	✓
31	3043	Isoorientin	6.84	C_21_H_20_O_11_	L1	512	449.1081 (+0.6)	569	447.0934 (−0.3)	329.0659 (100); 299.0552 (87); 353.0659 (34); 383.0763 (17); 413.0870 (15); 431.0977 (14)	327.0510 (100); 357.0616 (88); 447.0932 (51); 297.0408 (36); 285.0402 (24); 429.0830 (13)	[6,22]	✓	✓	✓	✓	✓
32	3014	Flavone base + 3O, 1MeO, *C*-Hex-Hex	6.96	C_28_H_32_O_16_	L2a	521	625.1762 (−0.2)	583	623.1612 (−0.9)	343.0816 (100); 313.0709 (87); 367.0812 (61); 427.1022 (38); 409.0916 (33); 445.1133 (33)	623.1602 (100); 341.0663 (39); 327.0510 (25)	[28]		✓			
33	3081	Isovitexin	7.02	C_21_H_20_O_10_	L1	528	433.1128 (−0.3)	593	431.0983 (−0.2)	271.0599 (100); 255.0654 (66); 295.0593 (47); 323.0550 (43); 311.0550 (41); 143.0340 (25)	311.0557 (100); 431.0977 (50); 283.0607 (17); 265.0513 (11); 255.0657 (9); 293.0450 (7)	[22]		✓			
34	3086	Loliolide	7.03	C_11_H_16_O_3_	L1	534	197.1174 (+0.9)	ND	ND	179.1066 (100); 133.1011 (88); 107.0858 (78); 91.0544 (63); 161.0958 (36); 197.1167 (23)	ND		✓	✓	✓	✓	✓
35	2990	8-*O*-methyl-7-hydroxyaloin A	7.16	C_22_H_24_O_10_	L2a	550	449.1441 (−0.3)	606	447.1299 (+0.5)	269.0811 (100); 254.0575 (40); 272.0681 (38); 287.0918 (31)	269.0458 (100); 327.0865 (59); 312.0642 (48); 284.0669 (26)	[29]		✓			
36	3079	Apigenin-*C*-hexoside-*O*-hexoside	7.24	C_27_H_30_O_15_	L2b	558	595.1658 (+0.1)	622	593.1512 (+0.1)	313.0707 (100); 283.0601 (50); 433.1131 (47); 337.0708 (42); 379.0814 (20); 397.0919 (18)	593.1512 (100); 293.0460 (78); 413.0883 (55); 311.0561 (8); 119.0356 (6); 473.1112 (5)	[30]	✓	✓	✓	✓	✓
37	2974	Apigenin-*C*- hexoside -*O*-pentoside	7.38	C_26_H_28_O_14_	L2b	578	565.1552 (+0.1)	640	563.1407 (+0.1)	313.0708 (100); 433.1131 (79); 283.0604 (55); 337.0708 (44); 415.1028 (24); 397.0920 (21)	293.0450 (100); 563.1393 (92); 413.0870 (45); 311.0553 (13); 341.0662 (9); 323.0556 (9)	[31]	✓	✓	✓	✓	✓
38	2991	Hydroxyaloin A ou B	7.39	C_21_H_22_O_10_	L2a	580	435.1285 (−0.2)	643	433.1140 (−0.1)	255.0654 (100); 227.0705 (28); 273.0760 (14); 85.0284 (5)	313.0721 (100); 270.0536 (92); 433.1143 (17); 284.0685 (8)	[22]	✓	✓	✓		✓
39	3000	Aloenin 2′-*p*-coumaroyl ester	7.47	C_28_H_28_O_12_	L2b	585	557.1655 (+0.3)	655	555.1507 (−0.2)	163.0388 (100); 275.0913 (63); 395.1339 (24); 557.1661 (21); 257.0803 (20); 299.0912 (16)	393.1191 (100); 273.0768 (61); 179.0349 (47); 135.0453 (25); 375.1078 (9); 245.0813 (8)					✓	
40	3212	Aloesin derivative	7.53	C_23_H_28_O_10_	L3	593	465.1757 (+0.4)	667	463.1612 (+0.5)	275.0912 (100); 233.0811 (43); 465.1755 (36); 345.1325 (19); 245.0808 (18); 257.0808 (17)	375.1086 (100); 243.0666 (20); 213.0556 (19); 255.0670 (19); 285.0772 (19); 87.0452 (17)					✓	
41	3076	Vitexin	7.54	C_21_H_20_O_10_	L1	595	433.1131 (+0.4)	670	431.0985 (+0.3)	255.0653 (100); 313.0709 (61); 283.0601 (29); 433.1126 (28); 227.0703 (23); 273.0758 (16)	311.0566 (100); 283.0614 (58); 431.0987 (46); 341.0672 (44); 323.0565 (13); 269.0456 (8)	[18]	✓	✓	✓	✓	✓
42	2194	Mirabijalone C	7.57	C_24_H_26_O_12_	L2b	599	507.1495 (−0.4)	ND	ND	285.0760 (100); 345.0976 (58); 165.0548 (30); 327.0873 (26); 267.0656 (20); 181.0496 (18)	ND			✓			
43	3217	Quercetin-*O*-hexoside	7.65	C_21_H_20_O_12_	L2b	611	465.1028 (+0.1)	687	463.0882 (0)	303.0499 (100); 85.0285 (11); 145.0498 (6); 127.0398 (6)	300.0287 (100); 463.0893 (51); 271.0259 (29); 255.0305 (15)	[22]	✓	✓			
44	3074	Vitexin pentoside +OCH_3_	7.77	C_27_H_30_O_15_	L2b	615	595.1659 (+0.3)	698	593.1515 (+0.5)	343.0812 (100); 313.0708 (61); 463.1242 (48); 367.0813 (46); 427.1027 (19); 409.0930 (18)	593.1524 (100); 323.0568 (72); 443.0987 (39); 341.0684 (11); 308.0328 (10); 371.0781 (8)		✓	✓	✓	✓	✓
45	3223	Vitexin-*O*-methyl	7.97	C_22_H_22_O_11_	L2b	645	463.1236 (+0.2)	729	461.1092 (+0.6)	343.0812 (100); 313.0705 (89); 367.0816 (36); 397.0928 (15); 427.1021 (12); 409.0923 (11)	341.0663 (100); 298.0479 (89); 461.1085 (52); 371.0758 (32); 353.0676 (6); 313.0723 (6)		✓			✓	✓
46	2258	Quercetin-*O*-malonylglucoside	8.08	C_24_H_22_O_15_	L2b	663	551.1032 (+0.1)	ND	ND	303.0496 (100); 127.0392 (22); 85.0288 (10); 109.0284 (9); 159.0306 (6); 145.0479 (5)	ND	[32]	✓				
47	3023	Coumaroylaloesin	8.21	C_28_H_28_O_11_	L2a	674	541.1702 (−0.4)	760	539.1558 (−0.2)	147.0442 (100); 275.0916 (38); 541.1705 (38); 395.1337 (16); 257.0811 (16); 299.0914 (10)	375.1087 (100); 163.0401 (63); 119.0502 (33); 255.0663 (16); 285.0769 (14); 243.0664 (13)	[25]		✓		✓	
48	3060	Kaempferol-3-*O*-rutinoside	8.23	C_27_H_30_O_15_	L2a	683	595.1654 (−0.6)	767	593.1513 (+0.2)	287.0552 (100); 85.0284 (9); 129.0546 (8); 147.0650 (2)	285.0397 (100); 593.1505 (92); 284.0324 (81); 255.0299 (4)	[20]		✓		✓	
49	3022	Mirabijalone C isomer	8.26	C_24_H_26_O_12_	L2b	687	507.1494 (−0.6)	772	505.1354 (+0.5)	285.0760 (100); 345.0973 (87); 327.0869 (45); 163.0391 (41); 303.0870 (14); 267.0655 (12)	343.0820 (100); 299.0922 (28); 505.1346 (20); 325.0714 (16); 257.0817 (15); 281.0815 (8)			✓			
50	3021	Hydroxyaloin A or B	8.29	C_21_H_22_O_10_	L2a	692	435.1286 (+0.1)	782	433.1143 (+0.6)	255.0653 (100); 227.0703 (25); 273.0758 (22); 85.0282 (5)	271.0609 (100); 313.0717 (4); 241.0503 (2)	[22]				✓	✓
51	3037	Astragalin	8.38	C_21_H_20_O_11_	L2a	699	449.1079 (+0.1)	791	447.0936 (+0.7)	287.0554 (100); 85.0283 (10); 127.0393 (5); 145.0497 (4)	447.0934 (100); 284.0327 (96); 255.0304 (65); 227.0353 (45)	[20]		✓			
52	3045	Narcissin or Narcissoside	8.46	C_28_H_32_O_16_	L2a	711	625.1765 (+0.3)	797	623.1615 (−0.4)	317.0658 (100); 85.0283 (10); 129.0545 (9); 147.0659 (3)	315.0502 (100); 623.1614 (72); 300.0275 (9); 299.0199 (9)	[20]				✓	
53	3042	Mirabijalone *C* isomer	8.54	C_24_H_26_O_12_	L2b	ND	ND	811	505.1351 (−0.1)	ND	343.0824 (100); 325.0719 (22); 299.0932 (18); 257.0823 (11)		✓	✓	✓		✓
54	3020	2″-*O*-feruloylaloesin or isomer	8.59	C_29_H_30_O_12_	L2b	728	571.1813 (+0.5)	819	569.1663 (−0.3)	177.0548 (100); 275.0916 (19); 571.1814 (17); 145.0286 (15); 395.1338 (9); 299.0907 (6)	375.1089 (100); 193.0507 (61); 134.0374 (39); 255.0664 (18); 285.0771 (14); 243.0666 (12)					✓	
55	3236	Isorhamnetin 3-*O*-glucoside	8.62	C_22_H_22_O_12_	L2a	742	479.1186 (+0.4)	833	477.1041 (+0.5)	317.0658 (100); 85.0281 (10); 145.0495 (6)	477.1040 (100); 314.0439 (68); 243.0293 (31); 271.0260 (30); 285.0404 (17)	[20]	✓				
56	3239	Isorabaichromone	8.73	C_29_H_32_O_12_	L2b	754	573.1964 (−0.4)	840	571.1822 (+0.2)	163.0392 (100); 217.0866 (65); 247.0968 (34); 349.1291 (21); 573.1972 (18); 409.1283 (14)	161.0245 (100); 571.1830 (37); 527.1564 (35); 553.1710 (24); 179.0349 (12); 135.0450 (10)	[33]		✓			
57	2371	Isoorientin + malonic acid moiety	8.93	C_24_H_22_O_14_	L2a	776	535.1077 (−0.9)	ND	ND	287.0546 (100); 127.0388 (21); 145.0493 (12); 159.0282 (12); 109.0291 (11); 85.0290 (8)	ND	[34]		✓			
58	2987	Aloin B	9.18	C_21_H_22_O_9_	L2a	795	419.1338 (+0.3)	913	417.1194 (+0.7)	239.0702 (100); 211.0753 (33); 257.0809 (29); 85.0282 (9)	297.0769 (100); 268.0744 (8); 251.0718 (4); 255.0656 (3)	[6]	✓	✓	✓	✓	✓
59	3255	2″-*O*-trans-*p*-coumaroylaloenin	9.37	C_28_H_28_O_12_	L2a	818	557.1655 (+0.3)	937	555.1509 (+0.2)	163.0387 (100); 275.0913 (91); 395.1338 (38); 233.0807 (18); 377.1230 (12); 299.0912 (9)	273.0770 (100); 393.1189 (60); 55.1503 (35); 303.0873 (15); 245.0818 (15); 179.0351 (12)	[6]	✓		✓		✓
60	3002	Isoaloeresin D	9.49	C_29_H_32_O_11_	L2b	830	557.2015 (−0.4)	950	555.1874 (+0.4)	147.0439 (100); 217.0857 (62); 557.2017 (54); 513.1751 (25); 247.0963 (25); 393.1330 (20)	145.0297 (100); 511.1614 (30); 555.1875 (26); 163.0403 (18); 117.0348 (13); 537.1771 (12)	[29]		✓			
61	2989	Aloin A	9.55	C_21_H_22_O_9_	L1	835	419.1338 (+0.3)	962	417.1193 (+0.5)	239.0703 (100); 211.0754 (50); 257.0810 (27); 85.0283 (8)	297.0765 (100); 268.0740 (8); 255.0667 (4); 251.0710 (4)	[6]	✓	✓	✓	✓	✓
62	3258	7-*O*-methylaloeresin A	9.65	C_29_H_30_O_11_	L2b	846	555.1859 (−0.3)	970	553.1718 (+0.5)	147.0439 (100); 259.0963 (43); 289.1068 (32); 555.1858 (26); 435.1432 (17); 313.1068 (15)	407.1342 (100); 145.0292 (60); 553.1706 (57); 163.0400 (40); 243.0659 (27); 119.0499 (16)	[29]		✓			
63	2982	Malonylnataloin or isomer	9.80	C_24_H_24_O_12_	L2a	873	505.1342 (+0.3)	989	503.1196 (+0.2)	239.0700 (100); 341.0649 (8); 109.0284 (7); 211.0756 (4)	297.0769 (100); 268.0743 (3); 459.1296 (2); 255.0663 (2)	[6]	✓	✓	✓	✓	✓
64	2998	5-hydroxyaloin A 6′-*O*-acetate	9.89	C_23_H_24_O_11_	L2b	886	477.1392 (+0.1)	1004	475.1245 (−0.2)	255.0654 (100); 313.0705 (24); 399.1073 (8); 273.0755 (9)	271.0614 (100); 313.0724 (3); 283.0618 (2)	[29]				✓	
65	2996	Coumaroylaloenin derivative	9.96	C_23_H_28_O_10_	L2b	894	465.1758 (+0.6)	1017	463.1610 (+0.1)	275.0914 (100); 233.0810 (44); 465.1754 (32); 299.0916 (26); 245.0808 (26); 257.0809 (21)	273.0765 (100); 463.1604 (87); 245.0813 (50); 375.1080 (41); 87.0451 (27); 231.0658 (23)		✓		✓		✓
66	3264	Aloesin coumaroyl hexoside	10.05	C_34_H_38_O_16_	L2b	905	703.2235 (+0.3)	1034	701.2088 (+0.1)	147.0442 (100); 275.0914 (50); 395.1338 (22); 541.1707 (21); 703.2230 (9)	701.2096 (100); 555.1730 (37); 285.0773 (31); 465.1413 (26); 163.0401 (22); 537.1619 (16)		✓		✓		✓
67	3265	Malonylnataloin	10.10	C_24_H_24_O_12_	L2a	912	505.1342 (+0.3)	1043	503.1195 (0)	239.0705 (100); 211.0757 (5); 109.0287 (4); 487.1243 (3); 281.0812 (3); 341.0658 (3)	297.0769 (100); 268.0740 (3); 459.1300 (3); 255.0663 (2)	[6]	✓	✓	✓	✓	✓
68	2988	Aloeresin A	10.20	C_28_H_28_O_11_	L2a	922	541.1708 (+0.7)	1052	539.1561 (+0.4)	147.0440 (100); 275.0915 (69); 541.1707 (23); 233.0809 (18); 395.1339 (17); 119.0491 (10)	273.0771 (100); 539.1568 (78); 393.1201 (45); 163.0402 (34); 375.1093 (26); 245.0823 (24)	[6]	✓		✓		✓
69	3007	4,2′,3′,4′-tetrahydroxychalcone 4′-*O*-(6″-*O*-*p*-coumaroyl)glucoside	10.30	C_30_H_28_O_12_	L2a	936	581.1652 (−0.3)	1066	579.1507 (−0.2)	147.0442 (100); 119.0493 (4)	313.0715 (100); 579.1510 (6)	[20]		✓			
70	3275	Aloeresin A +OCH_3_ on the coumaroyl moeity	10.51	C_29_H_30_O_12_	L2b	959	571.1813 (+0.5)	1100	569.1667 (+0.4)	177.0547 (100); 275.0914 (32); 145.0284 (12); 233.0809 (5)	273.0769 (100); 569.1673 (83); 193.0510 (44); 393.1192 (36); 375.1093 (30); 134.0374 (29)		✓		✓		✓
71	2555	3,4,5-trihydroxy-6-(hydroxymethyl)oxan-2-yl 3-(2-hydroxyphenyl)prop-2-enoate	10.52	C_15_H_16_O_7_	L2a	960	309.0972 (+1)	ND	ND	147.0445 (100); 119.0494 (21); 91.0549 (6); 165.0537 (7)	ND	[20]			✓		✓
72	3085	Feralolide	10.59	C_18_H_16_O_7_	L2b	969	345.0970 (+0.4)	1113	343.0823 (−0.1)	285.0761 (100); 163.0390 (83); 175.0390 (74); 327.0865 (49); 267.0652 (33); 123.0441 (20)	325.0727 (100); 343.0832 (74); 299.0936 (57); 257.0824 (53); 283.0621 (45); 173.0614 (37)	[35]	✓	✓	✓	✓	✓
73	3279	4,2′,3′,4′-tetrahydroxychalcone 4′-*O*-(6″-*O*-*p*-coumaroyl)glucoside	10.60	C_30_H_28_O_12_	L2a	975	581.1652 (−0.3)	1114	579.1506 (−0.3)	147.0443 (100); 119.0494 (5)		[20]		✓			
74	3001	Aloenin or Aloesin derivative	10.98	C_24_H_30_O_10_	L3	1002	479.1913 (+0.3)	1155	477.1767 (+0.2)	275.0916 (100); 479.1911 (50); 233.0809 (44); 299.0917 (25); 245.0810 (24); 377.1232 (15)	273.0774 (100); 477.1773 (91); 245.0825 (46); 375.1089 (40); 101.0610 (40); 231.0667 (22)		✓		✓	✓	✓
75	2980	Malonylnataloin derivative	11.67	C_23_H_24_O_10_	L3	1044	461.1440 (−0.5)	1225	459.1299 (+0.5)	239.0704 (100); 211.0754 (46)	279.0663 (100); 339.0873 (84); 251.0712 (67); 297.0758 (3)						✓
76	2652	Aloe C-glucosyl chromone	11.91	C_29_H_32_O_10_	L2b	1057	541.2065 (−0.6)	ND	ND	131.0492 (100); 541.2073 (73); 217.0862 (71); 497.1808 (38); 377.1385 (28); 247.0966 (25)	ND	[29]		✓			
77	2658	Chromone derivative (aloe glucosyl chromone)	12.05	C_29_H_30_O_12_	L3	1063	571.1806 (−0.7)	ND	ND	131.0490 (100); 103.0542 (8); 247.0959 (6); 571.1815 (3)	ND			✓			
78	3033	Microdontin A or B	12.29	C_30_H_28_O_11_	L2a	1079	565.1706 (+0.3)	1276	563.1561 (+0.4)	147.0438 (100); 119.0489 (5); 239.0694 (2); 91.0542 (2)	297.0775 (100); 563.1588 (3); 145.0304 (2)	[6]		✓	✓		
79	2994	Microdontin A or B	12.53	C_30_H_28_O_11_	L2a	1090	565.1706 (+0.3)	1296	563.1560 (+0.2)	147.0438 (100); 119.0492 (5); 239.0706 (2); 91.0546 (2)	297.0770 (100); 563.1552 (4); 268.0752 (3)	[6]		✓	✓		
80	2704	Isoeugenitin	12.96	C_12_H_12_O_4_	L2b	1109	221.0810 (+0.7)	ND	ND	221.0810 (100); 177.0547 (12); 91.0544 (9); 145.0649 (6); 115.0548 (5)	ND			✓	✓		✓
81	2758	Lysophosphatidylcholine (LPC) 18:3	18.42	C_26_H_48_NO_7_P	L2a	1163	518.3239 (−0.4)	ND	ND	184.0730 (100); 104.1068 (62); 86.0965 (14); 124.9991 (9); 518.3244 (6)	ND	[36]		✓			✓
82	2792	LPC 18:2	19.48	C_26_H_50_NO_7_P	L2a	1197	520.3398 (+0.1)	ND	ND	184.0733 (100); 104.1072 (55); 86.0963 (10); 124.9999 (8); 520.3380 (6)	ND	[37]	✓	✓	✓	✓	✓
83	3055	Lysophosphatidylethanolamine(LPE) 16:0	19.94	C_21_H_44_NO_7_P	L2a	1216	454.2929 (+0.2)	1466	452.2783 (+0.1)	313.2742 (100); 282.2794 (19); 216.0631 (14); 98.9841 (9); 155.0107 (7); 393.2351 (3)	255.2332 (100); 452.2789 (23); 256.2365 (14)	[20]	✓	✓	✓	✓	✓
84	2813	α-Linolelic acid	19.97	C_18_H_30_O_2_	L2b	1218	279.2319 (+0.5)	ND	ND	81.0697 (100); 95.0856 (96); 109.1013 (42); 123.1166 (27); 67.0541 (16); 137.1326 (15)	ND		✓	✓	✓	✓	✓
85	2818	LPC 16:0	20.08	C_24_H_50_NO_7_P	L2b	1223	496.3398 (−2)	ND	ND	184.0735 (100); 104.1071 (56); 86.0964 (8); 496.3399 (7); 124.9999 (7); 313.2734 (3)	ND	[20]	✓	✓	✓	✓	✓
86	3325	17-Hydroxylinolenic acid or isomer	20.36	C_18_H_30_O_3_	L2b	1231	295.2269 (+0.4)	1490	293.2121 (−0.4)	179.1434 (100); 99.0804 (73); 93.0696 (51); 135.1176 (47); 121.1016 (35); 277.2160 (20)	293.2125 (100); 89.0244 (45); 158.9778 (31); 227.0652 (24)	[20]	✓	✓	✓	✓	✓
87	2830	LPC 18:1	20.59	C_26_H_52_NO_7_P	L2b	1235	522.3555 (+0.2)	ND	ND	184.0731 (100); 104.1071 (60); 124.9995 (9); 86.0959 (8); 522.3542 (7); 258.1097 (3)	ND	[20]	✓	✓			✓
88	2847	LPC 17:0	20.97	C_25_H_52_NO_7_P	L2b	1252	510.3562 (+1.5)	ND	ND	184.0734 (100); 104.1072 (47); 86.0968 (8); 125.0013 (6); 510.3550 (5)	ND	[20]					✓
89	2856	LPC 18:0;O	21.84	C_26_H_54_NO_7_P	L2b	1261	524.3710 (−0.1)	ND	ND	184.0730 (100); 104.1073 (70); 125.0003 (8); 86.0957 (8)	ND	[20]	✓				
90	3066	α-linolenic or γ-linolenic acid	22.9	C_18_H_30_O_2_	L2b	1283	279.2320 (+0.5)	1538	277.2173 (−0.1)	95.0857 (100); 81.0695 (79); 109.1010 (76); 123.1168 (52); 137.1322 (16); 279.2318 (14)	277.2175 (100); 89.9254 (7); 218.0168 (6); 147.0441 (6)		✓	✓	✓		✓
91	3063	Linoleic acid	23.99	C_18_H_32_O_2_	L2a	1300	281.2476 (+0.3)	1555	279.2330 (+0.2)	97.1014 (100); 83.0854 (68); 111.1170 (52); 147.1163 (22); 121.1008 (25); 245.2259 (12)	279.2334 (100); 116.9267 (5); 100.9362 (4)	[20]	✓	✓	✓	✓	✓
92	3337	Pheophorbide A	24.73	C_35_H_36_N_4_O_5_	L2a	1316	593.2759 (+0.1)	1566	591.2610 (−0.5)	593.2762 (100); 533.2546 (13)	515.2457 (100); 500.2228 (11); 559.2361 (8); 471.2543 (5)	[20]	✓	✓	✓	✓	✓
93	3108	Pheophorbide A + CH_2_CH_2_ moiety	26.78	C_37_H_40_N_4_O_5_	L2b	1340	621.3071 (−0.1)	ND	ND	621.3067 (100); 561.2859 (17)	ND		✓	✓	✓	✓	✓

MN ID—MolNotator ID. IC—Identification confidence. MM ID ^+^—MzMine ID in positive ion mode. MM ID ^−^ —MzMine ID in negative ion mode. ND—Not detected. M—Aloe macra. V—Aloe vera. T—Aloe tormentorii. F—Aloe ferox. P—Aloe purpurea.

**Table 2 antioxidants-12-00050-t002:** Total phenolic content and antioxidant activity of leaf ethanolic extracts of the 5 *Aloe* species.

Species or Samples	TPC g GAE ^1^/100 g Extract	DPPH EC_50_ µg/mL (Mean ± SD)
*Aloe macra*	2.1 ± 0.0	172 ± 4 *
*Aloe vera*	1.5 ± 0.1	1340 ± 86 *
*Aloe tormentorii*	1.1 ± 0.0	902 ± 60 *
*Aloe ferox*	1.7 ± 0.0	151 ± 3 *
*Aloe purpurea*	2.6 ± 0.1	88 ± 1

^1^ Gallic Acid Equivalent; SD—Standard deviation; *—ANOVA followed by Dunnett’s multiple comparisons test results (*p* ≤ 0.05).

**Table 3 antioxidants-12-00050-t003:** Comparative quantitative analysis of antioxidant compounds using peak height in UV at 325 nm.

Compound	Peak Height (mAU) ± SD				
	*A. purpurea*	*A. ferox*	*A. macra*	*A. tormentorii*	*A. vera*
3-*O*-caffeoylquinic acid (5)	17 ± 1	12 ± 0	ND	ND	ND
Aloesin (10)	33 ± 2	75 ± 7	83 ± 1	26 ± 2	10 ± 1
4-*O*-caffeoylquinic acid (12)	15 ± 3	37 ± 3	ND	ND	1 ± 0
Luteolin-*C*-glucoside-*O*-pentoside (30)	85 ± 2	ND	51 ± 1	17 ± 2	24 ± 1
Isoorientin (31)	104 ± 2	50 ± 1	56 ± 1	6 ± 1	15 ± 2
N.D. *m/z* 949.2767	30 ± 2	ND	18 ± 0	ND	ND
Coumaroylaloesin (47)	ND	165 ± 3	ND	ND	5 ± 1
2″-*O*-trans-*p*-coumaroylaloenin (59)	57 ± 6	ND	51 ± 4	9 ± 0	ND
Aloeresin A (68)	445 ± 8	2 ± 1	524 ± 19	ND	8 ± 2

^1^ Positive control; N.D.—Non Determined; ND—Not Detected; SD—Standard deviation.

## Data Availability

The data presented in this study are openly available. Data is available in a publicly-accessible repository. Raw LC/MS data is available at: ftp://MSV000089679@massive.ucsd.edu (accessed on 13 November 2022) and treated LC/MS data at: https://doi.org/10.5281/zenodo.6884839 (accessed on 13 November 2022). GNPS Feature-Based Molecular Networking Jobs are available online at https://gnps.ucsd.edu/ProteoSAFe/status.jsp?task=62db3daa7df74ad8a273eb51fc27b3e0 (accessed on 13 November 2022) and at https://gnps.ucsd.edu/ProteoSAFe/status.jsp?task=9048ac276e6841d08a858c46088f0c40 (accessed on 13 November 2022) for positive and negative modes, respectively.

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
