# Peer review of "LC-MS Based Phytochemical Profiling towards the Identification of Antioxidant Markers in Some Endemic Aloe Species from Mascarene Islands"

_antioxidants, 2022, doi:10.3390/antiox12010050_

Round 1
Reviewer 1 Report
Comments:
The title could be shortened and precise. In my opinion, in this case “phytochemical profiling” is more appropriate than “phytochemical analysis”. Moreover, it is not necessary to listed the species names.
For example: “LC-MS based phytochemical profiling towards the identification of antioxidant markers in some endemic Aloe species”
Results and discussion: In depth discussion in the antioxidant part of the article is needed. The authors could compare their results with previously reported. Thus, the benefits of the proposed procedure will be highlighted.
Author Response
Comments:
The title could be shortened and precise. In my opinion, in this case “phytochemical profiling” is more appropriate than “phytochemical analysis”. Moreover, it is not necessary to listed the species names.
For example: “LC-MS based phytochemical profiling towards the identification of antioxidant markers in some endemic Aloe species”
Thank you for the proposal. We decided to modify the title of the publication to “LC-MS based phytochemical profiling towards the identification of antioxidant markers in some endemic Aloe species from Mascarene islands”
Results and discussion: In depth discussion in the antioxidant part of the article is needed. The authors could compare their results with previously reported. Thus, the benefits of the proposed procedure will be highlighted.
Thank you for your suggestion. The discussion has been integrated into the chapter 3.3.1. DPPH assay in 96-well plates”.
Reviewer 2 Report
The current manuscript (antioxidants-2065220) required major revision before to consider it suitable for publication in Antioxidants. Although the experimental design and the analytical platforms were well explained and adequate for the aim, the manuscript has several drawbacks that should be revised. Below the detailed comments
Why did not the authors have quantified the most abundant compounds based on available standard compounds? Indeed, these data could be strengthen what explained by On-Line HPLC DPPH assays
Table 1.
Compound 4 was assigned as dicaffeoylquinic acid. However, it is unusual its elution before 3-CQA and 5-CQA (J. Agric. Food Chem. 2007, 55, 929−936; Food Chemistry 129 (2011) 877–883; Rapid Commun. Mass Spectrom.2010;24: 2986–2992). Since a fragment ion at m/z 341 appeared upon MS/MS experiment, it could be related to caffeoyl-hexoside leading thus to assign the compound 4 as caffeoylquinic acid-hexoside rather than dicaeeoylquinic acid.
Peak 24. Feruloylquinic acid. Please check the product ion at m/z 93 if it is correct. Maybe it would be 193?
Compound 28. The authors signed it as Apigenin-6-C-glucoside-7-O-glucoside. However, it shares the same fragmentation behaviour of its isomer Apigenin-6-C-glucoside-4’-O-glucoside (J. Chromatogr. A 1161 (2007) 214–223) and authentic standard compound is needs to pinpoint the proper isomer. Thus, the compound 28 should be assigned as Apigenin-C-hexoside-O-hexoside.
Compound 30. Please detail the glycosidic linkage of pentoside sugar
Compounds 36-37. Based on MS/Ms fragmentation, both compounds seem to be apigenin-C-hexoside-O-hexoside (J. Chromatogr. A 1161 (2007) 214–223). Please check both compounds to reassign the proper identification.
Compound 43. The cited reference [15 in the manuscript] assigned it as isoquercitrin. No MS data are explained on putative isohyperoside assignment. Based on the fragmentation, compound 43 could be identified as quercetin-O-hexoside since both glucose and galactose moieties occurred in several plants containing quercetin skeleton. Finally, it seems that the positive and negative fragment ion should be changed in the column arrangement.
Compound 46 Orientin isomer + malonic acid. Since orientin has a [M+H]+ at m/z 449, by attaching a further malonyl group the [M+H]+ of malonyl orientin should be at m/z 535 rather than m/z 551. Flavonoids only occurred in plants as malonylglycosides and not as glycosides-malonic acid.
Compound 52 Narcissin. The authentic standard compound is needs to unambiguously identify it. Indeed, narcissin is Isorhamnetin 3-robinobioside that has a very similar fragmentation of its isomer narcissoside (Isorhamnetin 3-rutinoside), as reported in PubChem website.
Table 2.
The TPC values are expressed without SD unlike to what reported for DPPH assay. Moreover, no statistical analyses have been performed on DPPH values. Thus, the table 2 needs an extensive change to consider it suitable and useful to the manuscript.
Author Response
The current manuscript (antioxidants-2065220) required major revision before to consider it suitable for publication in Antioxidants. Although the experimental design and the analytical platforms were well explained and adequate for the aim, the manuscript has several drawbacks that should be revised. Below the detailed comments
Why did not the authors have quantified the most abundant compounds based on available standard compounds? Indeed, these data could be strengthen what explained by On-Line HPLC DPPH assays.
Dear Reviewer, thank you for your reply. The relative quantification has been integrated into chapter 3.3.2 On-Line RP HPLC DPPH assay. Thanks to this analysis, we have discovered that Aloin A, with the retention time of 9.55 min, initially reported by us as antioxidant, doesn’t seem to follow the correlation between the peak height of compound at 325 nm in active and inactive extracts. We found the same intensity of Aloin A peak at 325 nm in active and inactive extracts and couldn’t explain the radical scavenging activity. Therefore, we found that the compound, responsible for the radical scavenging activity is the peak which is very close to the peak of Aloin A, compound 59, 2″-O-trans-p-coumaroylaloenin, with the retention time of 9.37 min. The correlation of peak height at 325 nm in active and inactive extracts is well correlated with the radical scavenging activity.
Table 1.
Compound 4 was assigned as dicaffeoylquinic acid. However, it is unusual its elution before 3-CQA and 5-CQA (J. Agric. Food Chem. 2007, 55, 929−936; Food Chemistry 129 (2011) 877–883; Rapid Commun. Mass Spectrom.2010;24: 2986–2992). Since a fragment ion at m/z 341 appeared upon MS/MS experiment, it could be related to caffeoyl-hexoside leading thus to assign the compound 4 as caffeoylquinic acid-hexoside rather than dicaeeoylquinic acid.
Thank you for this correction. Additionally, the difference between the precursor ion 515.1406 and caffeoylquinicacid moiety 353.0893 of 162.051 corresponds well to caffeoylquinic acid-hexoside, than to dicaffeoylquinic acid with the theoretical difference of 162.033. We have modified this data in the table.
Peak 24. Feruloylquinic acid. Please check the product ion at m/z 93 if it is correct. Maybe it would be 193?
Dear reviewer, the fragment ion at 93.0343 is well present on MS2 spectra of compound 24.
Compound 28. The authors signed it as Apigenin-6-C-glucoside-7-O-glucoside. However, it shares the same fragmentation behaviour of its isomer Apigenin-6-C-glucoside-4’-O-glucoside (J. Chromatogr. A 1161 (2007) 214–223) and authentic standard compound is needs to pinpoint the proper isomer. Thus, the compound 28 should be assigned as Apigenin-C-hexoside-O-hexoside.
This compound has been annotated by comparison with the massbank data of Apigenin-6-C-glucoside-7-O-glucoside, however we agree that we cannot claim the presence of this compound without the injection of the authentic standard. We modified this entry in the table.
Compound 30. Please detail the glycosidic linkage of pentoside sugar
The pentoside is probably attached to the sugar part of the aglycone. However, the presence of an additional sugar may significantly change the retention time of the compound. Therefore we suggest the structure to be luteolin-C-glucoside-O pentoside and we have modified in the table.
Compounds 36-37. Based on MS/Ms fragmentation, both compounds seem to be apigenin-C-hexoside-O-hexoside (J. Chromatogr. A 1161 (2007) 214–223). Please check both compounds to reassign the proper identification.
Thank you for this suggestion. The compound 36 could be annotated as Apigenin-C-hexoside-O-hexoside according to Zengin et al., 2021 (Antioxidants. 2021 May 17;10(5):792). Compound 37 is supposed to be Apigenin-C-hexoside-O-xyloside, however, as we cannot affirm that sugar is xylose we suggest to change to Apigenin-C-hexoside-O-pentoside.
Compound 43. The cited reference [15 in the manuscript] assigned it as isoquercitrin. No MS data are explained on putative isohyperoside assignment. Based on the fragmentation, compound 43 could be identified as quercetin-O-hexoside since both glucose and galactose moieties occurred in several plants containing quercetin skeleton. Finally, it seems that the positive and negative fragment ion should be changed in the column arrangement.
We agree that we cannot claim the presence of isoquertirin as we observe the same loss for glucose and galactose moieties. Therefore, we modified this entry in the table to Quercetin-O-hexoside. Thank you for your attention, the positive and negative fragment ions has been inversed.
Compound 46 Orientin isomer + malonic acid. Since orientin has a [M+H]+ at m/z 449, by attaching a further malonyl group the [M+H]+ of malonyl orientin should be at m/z 535 rather than m/z 551. Flavonoids only occurred in plants as malonylglycosides and not as glycosides-malonic acid.
Actually, the corresponding structure in the reference is Quercetin-O-malonylglucoside. Thank you for your attention. The entry has been modified.
Compound 52 Narcissin. The authentic standard compound is needs to unambiguously identify it. Indeed, narcissin is Isorhamnetin 3-robinobioside that has a very similar fragmentation of its isomer narcissoside (Isorhamnetin 3-rutinoside), as reported in PubChem website.
We agree that the structures are very close and we modified this entry in the table.
Table 2.
The TPC values are expressed without SD unlike to what reported for DPPH assay. Moreover, no statistical analyses have been performed on DPPH values. Thus, the table 2 needs an extensive change to consider it suitable and useful to the manuscript.
Thank you for your suggestions. We have completed the TPC values by SD and DPPH results by statistical analysis.
Round 2
Reviewer 2 Report
Table 2. The SD values of TPC should have the same layout with the same decimal (i.e. ± 0.0 rather than 0). Moreover, remove gallic acid for both TPC and DPPH.
The manuscript can be accept after applying the required amendments
Author Response
Dear Reviewer,
Thank you for your review. Please find the modifications, indicated in violet in the revised manuscript.
With best regards,
Elnur Garayev